# MPLs-Pred: Predicting Membrane Protein-Ligand Binding Sites Using Hybrid Sequence-Based Features and Ligand-Specific Models

**DOI:** 10.3390/ijms20133120

**Published:** 2019-06-26

**Authors:** Chang Lu, Zhe Liu, Enju Zhang, Fei He, Zhiqiang Ma, Han Wang

**Affiliations:** 1School of Information Science and Technology, Northeast Normal University, Changchun 130117, China; 2Institute of Computational Biology, Northeast Normal University, Changchun 130117, China

**Keywords:** membrane protein, binding site prediction, protein-ligand, ligand-specific model

## Abstract

Membrane proteins (MPs) are involved in many essential biomolecule mechanisms as a pivotal factor in enabling the small molecule and signal transport between the two sides of the biological membrane; this is the reason that a large portion of modern medicinal drugs target MPs. Therefore, accurately identifying the membrane protein-ligand binding sites (MPLs) will significantly improve drug discovery. In this paper, we propose a sequence-based MPLs predictor called MPLs-Pred, where evolutionary profiles, topology structure, physicochemical properties, and primary sequence segment descriptors are combined as features applied to a random forest classifier, and an under-sampling scheme is used to enhance the classification capability with imbalanced samples. Additional ligand-specific models were taken into consideration in refining the prediction. The corresponding experimental results based on our method achieved an appreciable performance, with 0.63 MCC (Matthews correlation coefficient) as the overall prediction precision, and those values were 0.604, 0.7, and 0.692, respectively, for the three main types of ligands: drugs, metal ions, and biomacromolecules. MPLs-Pred is freely accessible at http://icdtools.nenu.edu.cn/.

## 1. Introduction

Membrane proteins (MPs) are an important type of protein along with soluble globular proteins, fibrous proteins, and disordered proteins [1]. MPs are involved in various crucial biological functions [2], such as transportation through membranes, immune system molecule recognition, hormone reception, etc., and this is why they have a strong potential to be the target of new drugs in the future, since over half of modern therapeutic drugs target MPs [3]. Therefore, exploring the functions of membrane proteins, especially their binding capability, remains to be of profound importance on various fronts, not least of which includes drug discovery [4]. Currently, commercial drug target discovery mainly relies on traditional approaches such as high throughput screening or functional assays, etc. However, more efficient and economical approaches are required to identify the MP-ligand binding sites (MPLs) in detail for modern medicine. Obviously, intelligent computation is a promising approach for this purpose.

During the past decades, many considerable efforts have been applied towards predicting soluble protein-ligand interactions, and these are classified into three major categories: structure-based, sequence-based, and hybrid methods that use both sequence and structure characteristics. Structure-based methods follow the assumption that proteins with similar functions always have similar global or local structures. As an example, identifying ligand binding sites by using structural modeling and information was widely researched [5,6,7,8]. Since there are fewer protein structures available for more extensive requirements, sequence-based methods have been used to directly predict the residues that probably interact with a particular ligand [9,10,11,12,13]. However, the performance of these methods depends highly on whether the sequence-derived features can describe the differences between binding and non-binding residues. Considerable attention has since been paid to the hybrid method that combines structural and sequential information [14,15,16,17]. Previous studies demonstrated that hybrid methods are often superior to others because they inherit the advantages from both structure- and sequence-based methods.

Suresh et al. [18] published the first sequence-based MPL predictor Tm-lig for membrane proteins. However, aside from this study, few peer-reviewed works have been reported, while plentiful attention had been paid to their soluble partners. It is obvious that the challenges remain in the membrane protein MPL prediction, but opportunities also currently exist.

The structure-based approach was not applied to membrane proteins directly in this paper since the 3D structures required for comparative homology predictions are not abundant for membrane proteins compared to soluble proteins, which is caused by the native environment [19,20,21] where the hydrophobic thickness of the lipid bilayer they are inserted into exists [22]. Consequently, sequence-based features are much more accessible than structure-based features for membrane proteins in many realistic scenarios. Thus, the sequence-based approach is preferred here rather than the structure-based prediction.

As in many typical computational biological modes of research, the imbalanced learning problem exists within MP-ligand binding prediction, where the number of majority samples (non-binding residues) is significantly larger than that of minority samples (binding residues). According to statistics, the number of non-binding residues is about 150–200 times more than of that of binding residues. Numerous studies have shown that directly using the traditional classifier algorithm to imbalanced problems often tend to bias toward the larger classes [23,24,25]. Therefore, the data imbalance phenomenon is an inevitable problem to be solved.

Most of the soluble protein-ligand binding site prediction methods, as well as the only-MPLs prediction method, focus on predicting universal ligand binging sites without considering the differences among various ligands. In fact, significant distinctions exist among the different types of ligands based on their size, structure, function, or other characteristics, and different types of ligands tend to attach with particular residues referenced in the surrounding environment. Many ligand-specific binding site predictors have been developed recently [26,27,28], and these are often superior to universal-purpose binding site predictors. Considering this, in addition to the universal ligand-binding residue prediction model, we further build ligand-specific models to predict drug-like compound-binding, metal-binding, and biomacromolecule-binding residues, respectively.

Finally, we analyze the characteristics of MPLs and the contribution of different types of features in detail. We find that the position-specific scoring matrix (PSSM) features are the most effective. At the same time, combining four types of features could more effective which achieve the highest MCC value. From a biological perspective, the PSSM, topology, evolutionary profiles (PSSM), topology structure (TOPO), physicochemical properties (PCP), and primary sequence segment descriptors (SeqSeg) present the evolutionary information, the fundamental structure, the microenvironment, and the original sequence composition of the ligand binding sites, respectively.

In this study, we developed a membrane protein-ligand binding site predictor MPLs-Pred. The limitations of the structure-based methods motivates us to extract discriminative features of the MPLs from sequence information alone, such as evolutionary profiles, topology structure, physicochemical properties, and primary sequence segment descriptors. To tackle the serious impact of the data imbalance phenomenon, a random under-sampling scheme was applied before using the random forest classifier to predict MPLs. The universal MPLs-Pred derived a 0.63 MCC (Matthews correlation coefficient) value on the independent validation test. Considering the distinction among the different types of ligand-binding residues, we divided the ligand-binding residues into three categories, including drug-like compound-binding, metal-binding, and biomacromolecule-binding residues. We built ligand-specific models to further improve the prediction performance and achieved consideracn the independent validation tests, respectively. MPLs-Pred is freely accessible at http://icdtools.nenu.edu.cn/.

## 2. Results and Discussion

### 2.1. Characteristics of Ligand-Binding Residues

In this work, the ligand-binding residue is coded by the features of evolutionary profiles, topology, physicochemical properties, and primary sequence segment descriptors. Before employing these characteristics as the feature spaces of the predictor, we demonstrated the effectiveness of each by statistical and experimental methods.

Figure 1 shows the relative composition of ligand-binding residues based on the universal training dataset and ligand-specific binding residues on the corresponding dataset. The relative composition might reflect the enrichment and distribution of binding residues. It is observed that polar hydrophilic residues such as cysteine (C), histidine (H), and aspartic acid (D) are more likely to be binding sites for all kinds of ligands, especially for metal-binding residues. Biomacromolecules prefer to interact with alkaline residues, such as histidine (H), arginine (R), and lysine (K), as well as the polar hydrophilic residues. This phenomenon also exists in drug-binding residues but is not as significant as others. The enrichment of metal- and biomacromolecule-binding residues is more obvious than that of drug-binding residues. It is probably because the drugs are manmade chemical compounds and not natural ones. The result of long-term evolution makes the proteins form a special region for recognizing specific natural ligands to interact with and perform specific functions. The phenomenon of the low specificity of drug-binding residues leads to the interaction of chemical drugs and membrane proteins that are not in a perfect one-to-one correspondence; this is why drugs always have side effects. It is obvious that different types of ligand-binding residues show a different preference for residue enrichment. Thus, the introduction of the ligand-specific predict strategy is explicable.

Furthermore, the residue preferences of the neighboring environment of target residues were investigated. Two-sample logos maps of universal ligand-binding residues against corresponding non-binding residues are shown in Figure 2. According to the illustration, the enrichment phenomenon of neighboring residues of the target is not remarkable, which is probably a reflection of the contribution limits of sequence neighbor residues during the interacting process. Thus, over-introducing the information of neighbor residues may cause noise. This phenomenon is different from the case of soluble proteins, and the detailed reason remains to be further explored. 

We further investigated the topology distribution of binding residues on the training dataset. According to the predicted result of HMMTOP, among the 10,143 binding residues, 7978 residues (about 79%) were located on the outer side of the membrane, 1788 residues (about 18%) were located on the inner side of the membrane, and only 377 residues (less than 4%) binding residues were located on the transmembrane region. The reason for this phenomenon is a special function of membrane proteins: transmitted ligand, signal, and energy inside and outside of the cell. The residues located between phospholipids are always stable in order to keep the channel structure of membrane proteins.

### 2.2. The Contribution of Features

As described in Section 3.2, “Selected Features”, we employed four kinds of features to construct the feature spaces of the predictor. To evaluate these features, we calculated the Pearson correlation coefficient between features and labeled the universal and ligand-specific training datasets. As shown by the heat map in Figure 3, the features of PSSM reveal the highest negative correlation with the label. The features of PCP also show significant correlations on all datasets. The features of TOPO and SeqSeg show a lower correlation with labels, which might be because the feature vector was too sparse. In a point-by-point comparison among the universal and ligand-specific datasets, the linear correlation between features and labels is relatively lower on drug-like compound data and significantly higher on metal data. This phenomenon is consistent with the experimental results.

We further analyzed the contribution of different kinds of features to the predictor. As shown in Table 1. The features of PSSM, which presents the evolutionary information of protein sequences, are the most effective. The sequence conservation is obvious in the binding regions, but not all the conserved sequenced segments contain binding sites. Thus, this feature would be a necessary condition for the prediction (seen from high ACC values) but is not a sufficient condition (seen from low Sen values). We found that the topology feature did not obviously improve the prediction performance for MCC, but it did highly increase the Sen. It means that the topology feature helps the predictor to discover the binding sites in the transmembrane segments, even though the predicted topology is not accurate enough. At the same time, four features can be more sufficient and balanced to exhibit the existence of binding sites with similarly high ACC and the highest Sen, which would lead to the highest MCC. From a biological perspective, topology can present the fundamental aspect of the structure of the transmembrane protein, PCP can present the microenvironment of binding regions, and SeqSeg can present the original sequence composition surrounding the binding sites. 

### 2.3. The Random Under-Sampling Scheme Influences Predictor Performance

Membrane protein-ligand binding site prediction is a typical imbalance problem. As illustrated in Table 1, the negative samples are about 140–200 times more than the positive ones and will cause considerable noise. In this study, we used the random under-sampling scheme to reduce the negative influence of imbalanced data. Due to the limited number of positive samples, we kept all of them and randomly selected some negative samples to build a sub-training dataset. The ratio of negative and positive samples in the sub-training dataset is the most important parameter that seriously affects the performance of the predictor. Figure 4 shows the tendency of the MCC value as the ratio changes on (a) the training dataset and (b) the independent testing dataset. We can see in Figure 4a that the predictor achieves the best MCC value when the ratio is at 1, decreases rapidly as the ratio increases, and tends to stabilize when the ratio is >30. We put forward an inference about this situation: When the ratio is <30, selected negative samples are too small to describe the distribution of the original negative samples, and this may cause serious information lost; when the ratio is ≥30, the MCC value tends to stabilize but decrease smoothly due to the increasing noise of redundant samples. The tendency of the MCC value from independent validation as shown in Figure 4b further verifies our inference. 

### 2.4. Comparison with other Machine Learning Methods Over Cross-Validation

In this section, we compare the random forest classifier with other classification methods on the training dataset. As illustrated in Table 2, it is obvious that the ensemble classifier, including AdaBoost and RF, performs better than others. This might be because of the serious imbalance between positive and negative samples, and the ensemble method can reduce the negative influence of imbalanced data. The random forest classifier achieves the best MCC value.

### 2.5. Performance of MPLs-Pred 

The performance of MPLs-Pred on the training datasets over 10-fold cross-validation test are listed in Table 3. By observing the illustration in Table 3 and compared with universal models, the ligand specific predictor can significantly improve the predictive effect for metal- and biomacromolecule-binding residues, especially for metal-binding residues. However, the prediction accuracy of drug-like compound-binding residues is much worse than the universal model. We have more interest in why drug-binding residue prediction performs worse than others. We speculate that drugs, which are man-made chemical compounds, are very different from natural ligands. According to the statistics from the previous section, the differences between drug-binding residues and non-drug-binding residues are not as significant as that of natural ligand’s. This is because the process of interaction between the membrane protein and the ligand is interventional: the membrane protein will not evolve a special region to recognize a particular drug and further its binding with it. Thus, features derived from sequences cannot commendably describe the characteristics of drug-binding residues.

In order to prove the robustness of the predictor, we further compared the universal model and ligand-specific models on the independent testing dataset. The details are illustrated in Table 4. We found that all the ligand-specific predictors performed better than the universal one. The metal-ligand binding sites predictor achieved the best performance because of the highly significant characteristics of metal-binding residues. 

### 2.6. Case Studies

To further demonstrate the effectiveness of the MPLs predictor on the universal model and ligand-specific models, we took a metal-binding (UniProt ID: P00959, PDB ID: 1pfu) protein and a drug-binding protein (UniProt ID: Q43133, PDB ID: 2j1p) in the testing dataset for case studies. 

P00959 is a cytoplasm membrane protein in *Escherichia Coli*. It is the target protein of ATP, tRNA, and zinc ions, and also participates in aminoacyl-tRNA ligase activity. Four zinc binding sites are annotated in UniProt. The protein structure and zinc ion-binding residues are visualized in Figure 5a. It is obvious that the four amino acids not contiguous in the sequence are spatially clustered, forming a functional domain. Furthermore, the prediction results generated by MPLs-Pred with the universal model and metal-specific model are also illustrated. The result shows that the metal-specific model outperforms the universal model. MPLs-Pred with the metal-specific model correctly predicted all four binding residues, and the universal model correctly identified three out of the four binding residues.

Q43133 is a chromoplast membrane protein of *Sinapis Alba*. It is the target protein of magnesium, isopentenyl diphosphate, and dimethylallyl diphosphate. Six dimethylallyl diphosphate binding sites are annotated in UniProt. The details of the protein structure and drug-binding model are visualized in Figure 5b. Similar to the previous example, the six binding residues also form a functional domain. MPLs-Pred also achieved considerable accuracy on this protein. The universal model correctly predicted three out of the six binding residues, and the drug-specific model correctly predicted five out of the six binding residues.

### 2.7. Comparison with Existing Methods on the Independent Testing Dataset

In this study, we proposed a novel method to predict ligand-binding residues in membrane proteins, named MPLs-Pred. In order to verify the generalization capability of MPLs-Pred, we compared it with existing predictors on the independent dataset. 

There is only one previous study on the field of membrane protein-ligand binding residue prediction created by M. Xavier Suresh and his team in 2015 [18]. This predictor is named Tm-lig and is freely accessed from http://tmbeta-genome.cbrc.jp/tm-lig/tm-lig.html. Tm-lig encodes the residues by PSI-BLAST-generated PSSM profiles and employes the Naïve Bayes classifier to predict the candidate ligand-binding residues in membrane proteins. We compared MPLs-Pred predictor with the Tm-lig predictor on the independent dataset. The details are illustrated in Table 5. 

The testing result proves that MPLs-Pred has better performance than Tm-lig on all the evaluation indexes, especially on the MCC value, which reflects the overall performance of the predictor.

### 2.8. Homo’s Membrane Protein-Ligand Interactions

We have much interest in *Homo’s* MP-ligand interactions. Here, we built an exclusive predictor to identify ligand-binding residues in *Homo’s* membrane proteins and achieved considerable performance with 10-fold cross-validation. The ACC, Spe, Sen, and MCC values were 0.992, 0.993, 0.705, and 0.486, respectively. 

In this section, we analyze the gene ontology and pathway of *Homo’s* drug-binding membrane proteins to help understand their functions.

Gene ontology (GO) is an important initiative to unify the representation of gene and gene product attributes across all species. The enrichment analysis is to test whether a GO term is statistically enriched for any given data. Figure 6 illustrates the GO analysis with up to 10 significantly enriched terms in (a) biological process (BP), (b) cell component (CC) and molecular function (MF), respectively. For 337 *Homo’s* proteins in the proposed dataset, 6100 biological processes are enriched and 4248 are statistically significant. Single-organism processes are the most important biological process for homo MPs. Six hundred and forty-two (642) cell components are enriched and 401 are statistically significant. Intracellular, cytoplasmic, and organelle are the top three CC enrichments, but they are not much better than others; membrane proteins are distributed in almost all organelles without significant difference. One thousand one hundred and fifty-three (1153) molecular functions are enriched and 561 are statistically significant. Binding with ligands is the most important function of membrane proteins.

KEGG (Kyoto Encyclopedia of Genes and Genomes) is a manually curated pathway database for understanding high-level functions and utilities of the biological system. The KEGG pathway is a collection of manually drawn pathway maps. The enrichment analysis of the KEGG pathway helps researchers understand the pathway a given set of proteins are involved in. For *Homo’s* ligand-binding MPs, enriched processes are shown in Figure 7.

### 2.9. Performance Evaluation

The proposed prediction model MPLs-Pred was first evaluated by 10-fold cross validation on the training dataset. First, positive samples and negative samples were randomly divided into 10 equal parts, respectively. Then, one part of the positive and negative subset was chosen to build the validation dataset, and the remaining samples were used for training. This process was repeated ten times to build ten sub-predictors. The final performance is the average value of the sub-predictors. Then, independent validation was used to evaluate the generalization of the proposed method. Four metrics were employed to evaluate the performance of the predictor—specificity (Spe), sensitivity (Sen), accuracy (ACC), and the Matthews correlation coefficient (MCC):
$$\mathrm{Spe}=\frac{TN}{TN+FP}$$
$$\mathrm{Sen}=\frac{TP}{TP+FN}$$
$$\mathrm{ACC}=\frac{TP+TN}{TP+TN+FP+FN}$$
$$\mathrm{MCC}=\frac{TP\times TN-FP\times FN}{\sqrt{\left(TP+FP\right)\times \left(TP+FN\right)\times \left(TN+FP\right)\times \left(TN+FN\right)}}$$
where TP, FP, TN, and FN represent true positive, false positive, true negative and false negative, respectively.

The prediction of membrane protein-ligand binding residues is a typical imbalanced learning problem where the number of non-binding residues is significantly more than that of binding residues. Thus, excessive pursuit of overall accuracy is one-sided. In the field of practical application, researchers always expect that the predictor can provide more accuracy of ligand-binding residues instead of non-binding ones in order to obtain more candidate targets. In view of this, the MCC value, which provides an overall measurement of performance of binary classification problems, is regarded as the most reliable evaluation index in the experiments of this paper. The performance of the predictor is positively correlated with the MCC value.

## 3. Materials and Methods

### 3.1. Benchmark Datasets

A dataset of membrane protein-ligand binding sites had been extracted from the Protein Data Bank [29] by Suresh et al. [18] in 2015. This dataset contains 42 non-redundant protein sequences with 10,657 residues, and 1431 of them are identified as binding residues. Considering the dramatically increasing number of membrane proteins in recent years, we constructed a new benchmark dataset. First of all, we analyzed the annotations of all 102,429 manually annotated and reviewed membrane proteins from the Universal Protein Resource Databank (UniProt) released to date. Then, after removing protein sequences of less than 50 residues in length, as well as those with unknown residues such as X, we are left with 17,590 sequences with exact ligand-binding residues. To reduce the influence of data redundancy and homology bias [30], these proteins were clustered by CD-HIT with a 30% sequence identity cut-off, and the representative sequence in each cluster was picked. After that, we were left with 2734 proteins with 10,979 binding residues. To evaluate the effectiveness of the proposed method, these proteins were divided randomly into a training dataset with 2500 proteins and an independent validation dataset with 234 proteins. The data can be found in Appendix A: UniversalData.zip.

The protein binding ligand can be roughly divided into the following major categories: drug-like compounds, metal ions, and biomacromolecules, among which biomacromolecules include proteins, fats, sugar, nucleotides, and so on; thus, we divided the dataset into three parts according to the type of ligand. Other types of ligands are ignored because the sample size is too small to be statistically significant. The details of the universal dataset and the ligand-specific datasets are illustrated in Table 6. The ligand specific data can be found in Appendix A: LigandSpecificData.zip.

### 3.2. Selected Features

To distinguish ligand-binding and non-binding residues, we chose four kinds of features that can describe the characteristic of ligand-binding residues in membrane proteins, which always have a specific residue composition, evolutionary conservation, physicochemical environment, or topology characteristic. In this study, we employed the sliding window scheme to express the influence of the neighboring residue. Here, we set the value of window size to 7 residues, in which 3 were from upstream and 3 were from downstream, after testing multiple values.

#### 3.2.1. Evolutionary Profiles (PSSM)

It has been proven that highly conserved regions are always involved in basic cellular function. Research on the membrane [18] and non-membrane [11] protein-ligand interacting residues indicates that this evolutionary information is useful. Position-specific scoring matrix (PSSM) has been demonstrated as an effective feature for encoding the evolutionary information of protein sequences. It was widely used in many bioinformatics problems such as protein function prediction [31], membrane protein-protein interaction sites prediction [32], protein secondary structure prediction [33], DNA-binding proteins prediction [34,35], etc. 

For a protein sequence with L residues, we obtained its PSSM using PSI-BLAST [36] to search the non-redundant database (ftp://ftp.ncbi.nlm.nih.gov/blast/db/nr.tar.gz) through 3 iterations with a 0.001 *E*-value cutoff. An L×20 matrix was generated for each protein. Then, a sliding window was used for each residue to build its PSSM feature vector. Since the window size was 3, the dimensionality of the PSSM feature was 7×20=140 for each residue. 

#### 3.2.2. Topology Structure (TOPO)

Membrane proteins span or partly span the lipid bilayer of the membrane, which is the major difference between MPs and non-MPs. Thus, the fundamental aspect of the structure of the transmembrane protein is the membrane topology, i.e., the number of transmembrane segments, their position in the protein sequence, and their orientation in the membrane [37,38]. The topology structure features have been applied in various membrane protein-associated studies, such as helix-helix interactions and residue contacts [8,39]. Many researchers pay attention to predicting the topology of the transmembrane protein, and the newest predictor can achieve an accuracy value close to 80%. 

In this work, the topology of transmembrane was predicted by TOPCONS [40], where a reliable predictor marks each residue in the sequence as I, O, M or U, which represents the residue located on the inside, outside, membrane region, or non-membrane region but location unknown, respectively. We further digitized the topology descriptor with a 3-dimension vector-ignoring U. Each element in the vector represents the count of the corresponding topology in the sequence window. Hence, we obtained topology features in 3-dimension.

#### 3.2.3. Physicochemical Properties (PCP)

Since residues are fundamental building blocks of proteins, their physicochemical properties (PCP) influence the microenvironment of proteins, including energy, surface motions, dynamics, and so on [41]. Previous studies have shown that PCP can be used in many prediction methods such as enzymatic protein identification [42] and protein lysine acetylation prediction [43], etc. Previous studies show that physicochemical properties play important roles in the success of soluble protein-ligand binding site prediction. Since MPLs always appear in the water-soluble regions in membrane proteins, PCP can also be used in the field of MPL prediction.

In this study, 15 PCPs were collected from AAindex [44] based on research experience, named hydrophilicity value, hydrophobicity, net charge, polarity, size, residue volume, molecular weight, diameter, amino acid composition, composition of amino acids in membrane proteins, side chain interaction parameter, solvation free energy, transfer free energy, average flexibility indices, and accessible surface area. A further experiment showed that 15 PCPs can significantly improve the performance of the predictor. Thus, a 15×7=105-dimension vector was formed to represent the physicochemical properties of each residue.

#### 3.2.4. Primary Sequence Segment Descriptors (SeqSeg)

The original sequence is very important in that it can directly decide the structure and function of the protein; peptides with particular functions always show special arrangements. According to a study of the compositional difference between ligand-binding and non-ligand-binding residues, we built a 20-dimension vector to elucidate the primary sequence segment around the target residue. The value of the element in the vector represents the number of the corresponding residue in the protein segment sliced by the window. Thus, we obtained a 20-dimension vector to represent the features of the primary sequence segment around the target residue.

Finally, the feature space of the target residue was  a 140+4+105+20=269-dimension vector which contained four different kinds of sequence-derived features. The analysis of these features are be further described in Section 2.2, “The Contribution of Features”.

### 3.3. Random Under-Sampling (RUS)

The statistics in Table 1 illustrate that the MPL prediction problem is a typical imbalanced learning problem where the number of majority samples (non-binding residues) is significantly larger than that of minority samples (binding residues). The data imbalance phenomenon in the MPL prediction problem is very serious: The number of non-binding residues is about 140–200 times that of the binding residues. Numerous studies have shown that directly using the traditional classifier algorithm to imbalanced problems often tend to bias toward the larger classes [23,24,45,46]. Therefore, the data imbalance phenomenon is an inevitable problem to be solved. In this study, a Random under-sampling (RUS) scheme was applied before training the predictor to reduce the negative influence of imbalanced data. Since there is a limited number of binding residues, we contained all binding residues and randomly selected 30 times the number of non-binding residues. The impact of the RUS scheme on the predictor is further discussed in Section 2.3, “The Random Under-Sampling Scheme Influences Predictor Performance”. 

### 3.4. Random Forest Classifier 

Random forest (RF) is a powerful classifier first proposed by Leo Breiman in 2001 [47]. It is widely used for classification, regression, feature selection, and other tasks in the field of bioinformatics [48,49], such as prediction of protein-protein interaction sites [50], identification of membrane protein types [51], prediction of GPCR-drug interactions [52,53], and so on. RF is an ensemble of decision trees that add an additional layer to bagging them together. Decision trees in RF are trained by a subset randomly selected from the primary feature set of data, which is also randomly under-sampling the training dataset. After obtaining the forecast results, these decision trees vote on the class for the given input sample. In this study, the predictor showed the best performance when the CART split scheme was used with the number of trees set at 140 and the dimension of the candidate at each split set at 20. Because the features and training samples of each decision tree are randomly selected, random forests can handle imbalance samples with high-dimension well without sampling or feature selection processes. 

## 4. Conclusions

In this paper, we propose a novel membrane protein-ligand binding residue predictor named MPLs-Pred. We predicted the target residue using four types of sequence-derived features including evolution profiles, topology structure, physicochemical properties, and primary sequence segment descriptors. Then, the random forest classifier was employed to predict if a given residue is a ligand binding site or not. Experimental results showed that MPLs-Pred achieves considerable performance with MCC values of 0.597 and 0.356 on cross-validation and independent validation, respectively. Above this, we propose ligand-specific models that classifies ligands into drug, metal, and biomacromolecule to further improve the prediction accuracy. The ligand-specific models significantly improve performance compared with the universal model.

## Figures and Tables

**Figure 1 ijms-20-03120-f001:**
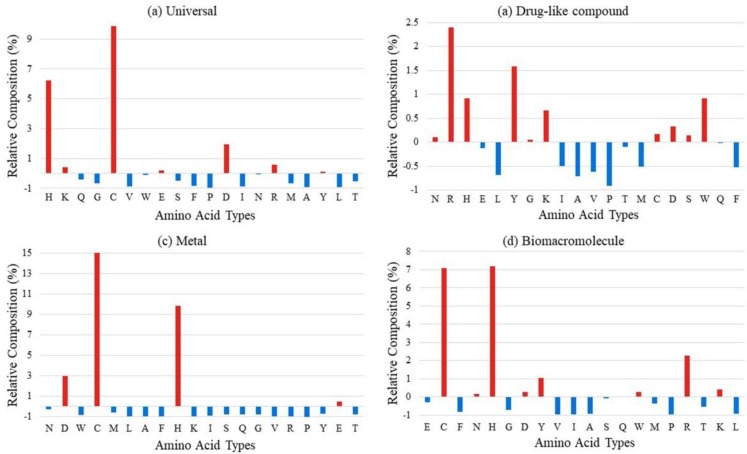
The relative composition of (**a**) universal ligand-binding residues, (**b**) drug-like compound-binding residues, (**c**) metal-binding residues, and (**d**) biomacromolecule-binding residues based on background distribution of all residues in corresponding datasets.

**Figure 2 ijms-20-03120-f002:**
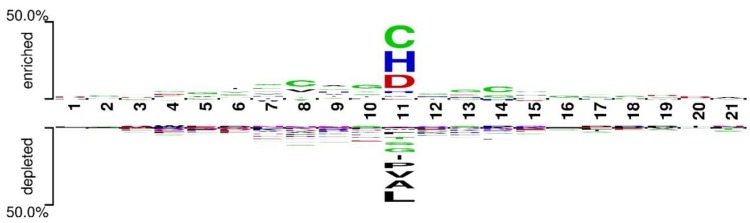
Two-sample logos of universal ligand-binding residues against non-binding residues.

**Figure 3 ijms-20-03120-f003:**
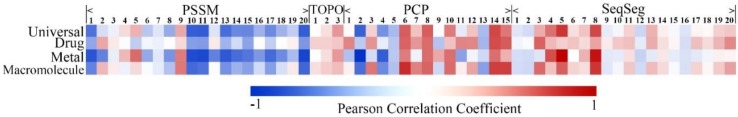
The heat map of the Pearson correlation coefficient between features and labels. Red represents positive correlations and blue represents negative correlations. The darker the color, the higher the correlation.

**Figure 4 ijms-20-03120-f004:**
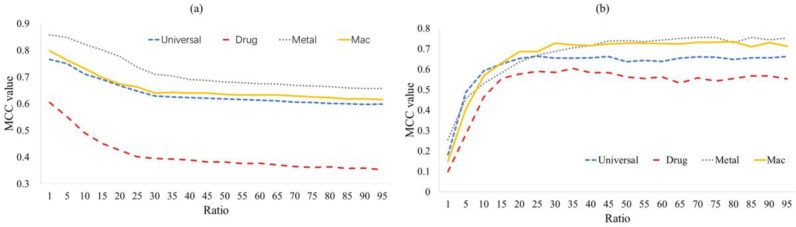
The tendency of MCC value as the ratio of non-binding residues and binding residues increase over (**a**) 10-fold cross-validation test and (**b**) independent validation test.

**Figure 5 ijms-20-03120-f005:**
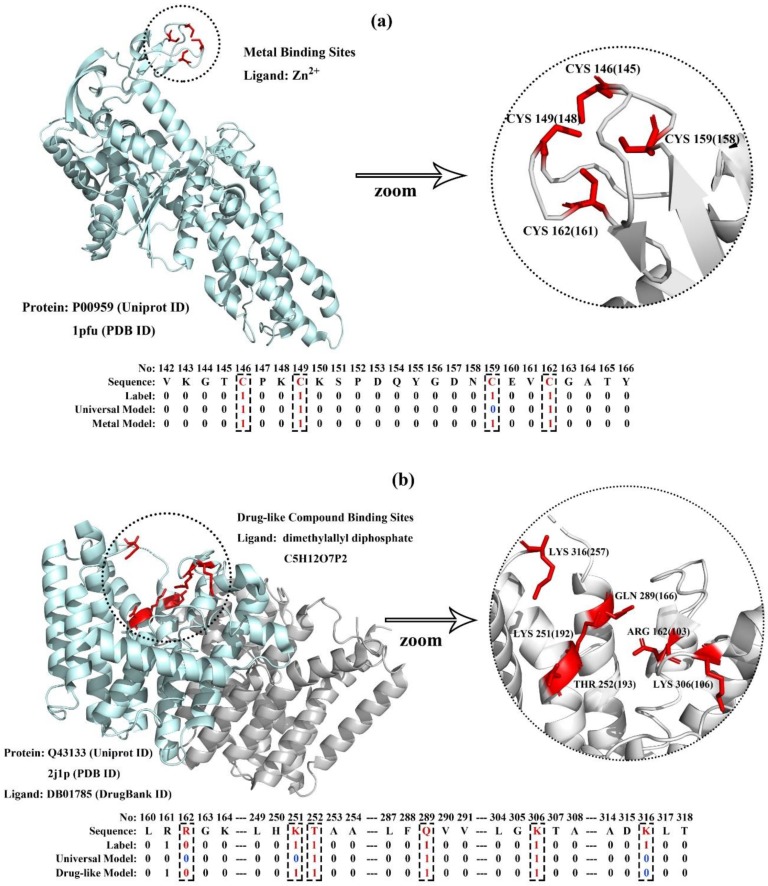
Visualization of membrane protein-ligand binding sites and their corresponding prediction result generated by MPLs-Pred with the universal model and ligand-specific models. As examples: (**a**) P00959: metal-binding MP with four zinc ion-binding residues, and (**b**) Q43133: drug-binding MP with six dimethylallyl diphosphate-binding residues.

**Figure 6 ijms-20-03120-f006:**
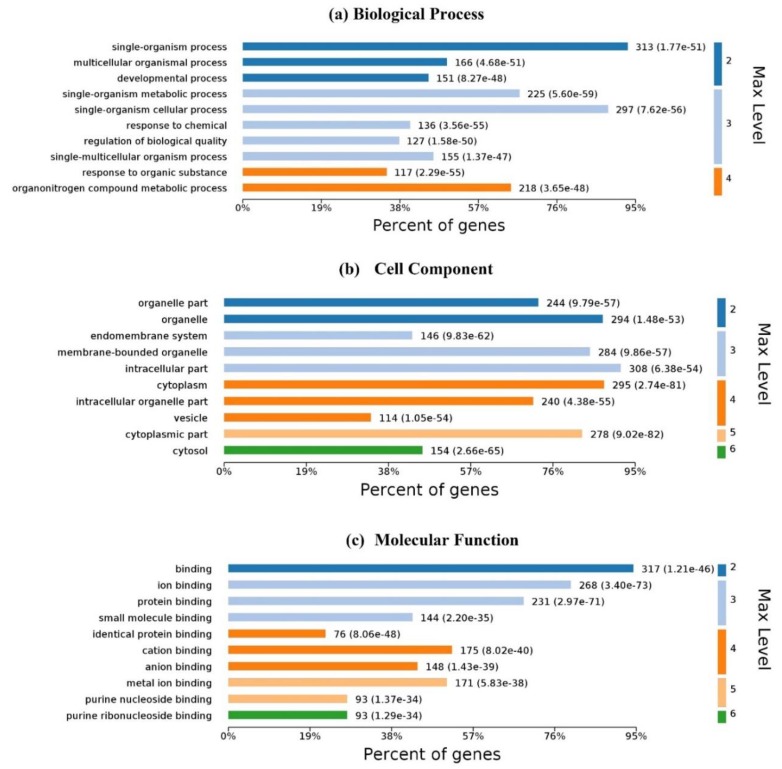
Gene ontology enrichment of *Homo’s* ligand-binding proteins: (**a**) biological process enrichment; (**b**) cell component enrichment; (**c**) molecular function enrichment. The graphs were made with OmicsBean.

**Figure 7 ijms-20-03120-f007:**
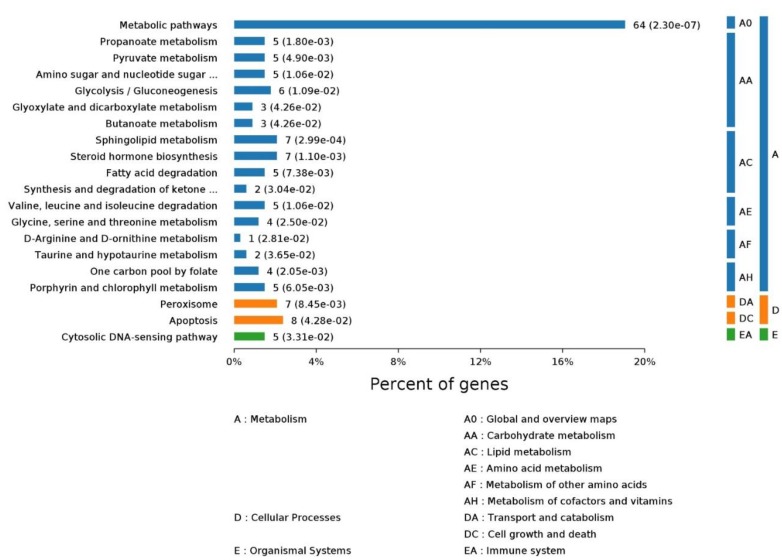
The KEGG (Kyoto Encyclopedia of Genes and Genomes) pathway enrichment of *Homo’s* ligand-binding proteins. The pictures were made with OmicsBean.

**Table 1 ijms-20-03120-t001:** The performance of different combinations of features.

Feature Combination	ACC	Spe	Sen	MCC
PSSM	**0.984**	**0.999**	0.386	0.582
TOPO	0.548	0.309	0.788	0.113
PCP	0.973	0.998	0.216	0.413
SeqSeg	0.973	0.999	0.189	0.389
PSSM+TOPO	0.98	0.998	0.415	0.603
PSSM+PCP	0.979	0.998	0.421	0.599
PSSM+ SeqSeg	0.98	0.998	0.415	0.601
TOPO+PCP	0.973	0.998	0.22	0.416
TOPO+SeqSeg	0.904	0.981	0.139	0.201
PCP+SeqSeg	0.973	0.998	0.213	0.41
PSSM+TOPO+PCP	0.979	0.998	0.422	0.6
PSSM+TOPO+SeqSeg	0.98	0.998	0.416	0.603
PSSM+PCP+SeqSeg	0.979	0.998	0.421	0.601
TOPO+PCP+SeqSeg	0.973	0.998	0.216	0.414
PSSM+TOPO+PCP+SeqSeg	0.971	0.997	**0.464**	**0.627**

Bold text in the table indicates that the feature achieved the highest value on this evaluation index.

**Table 2 ijms-20-03120-t002:** Comparison of random forests (RF) with other classifiers.

Method	ACC	Spe	Sen	MCC
SVM	0.9578	0.98	**0.774**	0.347
Naïve Bayes	0.844	0.85	0.67	0.246
AdaBoost *	**0.974**	0.995	0.334	0.472
RF	0.971	**0.997**	0.464	**0.627**

* Adaboost classifier-employed decision tree as the basic classifier. Bold text in the table indicates that the feature achieved the highest value on this evaluation index.

**Table 3 ijms-20-03120-t003:** Performance of the membrane protein-ligand binding site predictor (MPLs-Pred) on the training dataset with the universal model and ligand-specific models over 10-fold cross-validation.

Model	ACC	Spe	Sen	MCC
Universal	0.971	0.997	0.464	0.627
Drug	0.973	1.0	0.153	0.366
Metal	0.984	0.997	0.589	0.704
Biomacromolecule	0.936	0.993	0.481	0.629

**Table 4 ijms-20-03120-t004:** Performance of MPLs-Pred on the independent testing dataset with the universal model and ligand-specific models.

Model	ACC	Spe	Sen	MCC
Universal	0.996	0.998	0.618	0.63
Drug	0.997	1.0	0.397	0.604
Metal	0.996	0.998	0.759	0.7
Biomacromolecule	0.997	0.999	0.596	0.692

**Table 5 ijms-20-03120-t005:** Comparison of MPLs-Pred with the previous study on the independent dataset.

Method	ACC	Spe	Sen	MCC
Tm-lig	0.857	0.857	**0.769**	0.126
MPLs-Pred	**0.996**	**0.998**	0.618	**0.63**

Bold text in the table indicates that the feature achieved the highest value on this evaluation index.

**Table 6 ijms-20-03120-t006:** The detailed composition of new built standard datasets.

Dataset	Training Dataset			Testing Dataset		
	No. of Proteins	No. of Residues	Ratio ^1^	No. of Protein	No. of Residues	Ratio ^1^
Universal	2500 ^2^	(10,143, 1,524,372)	1:150	234	(836, 164,792)	1:197
Drug	655	(1839, 386,979)	1:210	45	(121, 25,193)	1:208
Metal	1375	(5734, 804,610)	1:140	117	(503, 85,437)	1:170
Biomacromolecule	857	(2505, 435,298)	1:174	67	(161, 35,022)	1:218

^1^ Figure in Ratio represents the ratio of binding to non-binding residues. ^2^ There are overlaps among datasets since some proteins interact with two or more type of ligands.

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
