# Peer review of "MPLs-Pred: Predicting Membrane Protein-Ligand Binding Sites Using Hybrid Sequence-Based Features and Ligand-Specific Models"

_ijms, 2019, doi:10.3390/ijms20133120_

Round 1

Reviewer 1 Report

The authors of the manuscript “MPLs-Pred: Predicting Membrane Protein-Ligand Binding Sites Using Hybrid Sequence-based Features and ligand-specific models” present an economical in silico approach for identifying potential residues in membrane proteins responsible for ligand binding, as well as classifying the possible ligands into three biochemical groups (Metal, Drug-like, Macromolecule). The algorithm is also made publicly available to researchers.

Overall, the work is significant and the conclusions are supported by the presented data. The authors convincingly show the performance of their method surpasses that of the other currently available method ((Tm-lig). I believe the work is relevant and of interest to the community and should be published after some minor revisions and recommendations:

1-     The main problem as the manuscript stands is redaction issues regarding the English language. Besides numerous single-word typos in the text, the overall English grammar is confusing along the entire manuscript and leaves the reader with extra mental work required to understand the beautiful and valuable message the authors intend to cover. It would be of great benefit if the authors could recruit specialized help to aid in the writing the manuscript.

2-     It would be helpful if the authors could elaborate at least minimally regarding how the problem of ligand binding site prediction for membrane proteins differs from that of soluble proteins. Is it possible to use soluble protein prediction algorithms with membrane proteins ? Yes/No and why ?.  The first impression is that this could be possible given that most ligands bind membrane proteins at their soluble/ solvent exposed portions, therefore the problem doesn’t seem different from the case of soluble proteins. Maybe the authors could elaborate a bit in this regard as a way to fundament the importance of their approach.

3-     Although it escapes the scope of the manuscript, it could be interesting to discuss at least, the case where ligands are embedded in the lipid bilayer (i.e. lipids such as sterols) and module the membrane protein function by binding from the hydrophobic core of the membrane into transmembrane regions, rather than the typical soluble portions mentioned above.  This problem has not been addressed in silico that I know of and it would be great if the authors could discuss it at least as a potential future venue to follow. 

Author Response

We upload the response as a file named 'Response to Reviewer1.pdf'

Reviewer 2 Report

The manuscript “MPLs-Pred: Predicting Membrane Protein-Ligand Binding Sites Using Hybrid Sequence-based Features and ligand-specific models” submitted by Chang Lu et al. describes a new membrane protein-ligand (MPLs) predictor. The predictor combined evolutionary profiles, topology structure, physicochemical properties and primary sequence segment descriptors. An under-sampling scheme is used to enhance the classification capability with imbalance samples. Then, the prediction is refined by additional ligand-specific models. Overall, the manuscript is well written, organized and the method of prediction proposed is potentially interesting for the field. 

Major comments:

1)The link to the MPLs-Pred server provided by the authors is not functional. After loading the data, the server is sending an empty mail without any attachment or neither analysis.  Please provide a functional server in order to evaluate the predictor. 

2)Regarding this point, and related to my previous comment I would ask the authors to provide the analysis for the following test samples: 

-Case I- Uniprot ID: O75976  

-Case II- Uniprot ID: Q99720

-Case III- Uniprot ID: Q9H6L5

Minor comments:

3)Lines 231-234-Figure 1. Please explain the color code (i.e Red and blue) used in this figure. Glutamic acid (E) can coordinate and bind metals. According to the figure, this amino acid does not appear as a metal binding residue? Please explain and discuss this in the text.

4)Line 353- Please correct the mistake in the name of the first author from ref. nº 18.  

Author Response

We upload the response as a file named 'Response to Reviewer2.pdf'

Reviewer 3 Report

The manuscript describes a web based tool, MPLs-Pred for the prediction of membrane protein-ligand binding sites, which uses protein sequence to predict whether a given amino acid residue in membrane protein sequence is a ligand binding residue or not. The article allows a potential useful feature in general to this field. There are a number of minor grammatical errors in the text that need to be fixed, as they detract from article as a whole; For example phrases like ‘Line- 45-47: the sequence-derived features could maximumly impact the spatial structural properties of the protein. ’ are grammatically incorrect and somewhat confusing. 

My main concern is that the server is not working; I got an error when submitting a protein sequence to the web server: 404 not found. Moreover, the web server is not well documented, there are no instruction, no example input or output neither a collection of useful biological examples/results. The web server must be running smoothly with full documentation before further consideration of the manuscript.

Some minor changes:

Please add references for the prediction of protein ligand interactions- Line38-40. 

Line 40-43 -"Structure-based methods follow the assumption that ligand always interacts with those proteins with similar structural properties in global or local.” the sentence is not clear to me what do you mean by similar structural properties in global or local. Can you expand it or rephrase it? 

Line 51 "el al." should be italics.

Table 1 footnote- Is it necessary to write 2-tuple?

line 164- "PCP could also be used in the filed of MPLs prediction." it sholud be field here.

Please rephrase Figure 5 legend; it is confusing. one of sentence starts with “And” here.  

Line 328: Escherichia Coli; it should be Escherichia coli in italics.

Author Response

(The authors gave the same response as above.)

Reviewer 4 Report

The manuscript reports development of a method for predicting ligand binding sites in membrane proteins. The manuscript suggests that prediction of ligand binding sites for membrane proteins is a separate problem. However, I think that many of the existing methods are applicable to both soluble and membrane proteins. Many of the leading methods use model structures so despite the limited number of membrane proteins in the Protein data bank it should still be possible to make predictions for many membrane proteins.

Further:

1.The introduction states that the PSSM features are the most effective - it is therefore difficult to see how this is membrane protein specific. Indeed the only features that are membrane related are the topology predictions from TOPCONS and from Figure 3 it is clear that topology is not highly informative as a feature. 

2. The English language needs to be improved.

3. The introduction misses an important set of methods that use protein structural modelling and information from related proteins structures to predict ligand binding sites. These include FINDSITE (Skolnick 2007), 3dligandsite (Sternberg 2010) and FunFold/IntFOLD (McGuffin L 2011).

4. I do not understand the following statement "As many typical computational biology research, MP-ligand binding prediction will also exist 61 the imbalanced learning problem, where the number of majority samples is significantly larger than 62 that of minority samples” What are majority and minority samples? This needs to be clearer, I assume they are referring to the residues that form a binding site are small in number.

5. The number of proteins in some parts of the test set are small (i.e. for the drug-like and biomarcromolecule sites), does this cause problems with overfitting? Also is there much difference between these two groups - it may be better to consider them together?

6.The testing of the method uses accuracy and specificity and sensitivity, given the imbalanced set I am pleased to see that MCC is also used, however it would be much better to remove the first measures and instead use precision, recall and F1 scores. For example it seems the PSSM is able to correctly identify nearly all of the residues that do not bind ligands but correctly identifies a much smaller proportion of the actual binding residues (Table 2).

7. The use of these different metrics would also help understand the results in table 3 where the SVM approach obtains high sensitivity and specificity but much lower MCC thank the random forest.

8. Figure 5 - avoid use of red and green (cannot be seen by those that are colour blind)

9. While the results suggest that the method here outperforms another method (tm-lig), it is not clear that this is a fair comparison given that tm-lig was trained on only 42 proteins, compared to the much larger training set used here.

10. What is a “homo protein”.  I do not see the relevance of the GO term analysis - how does this relate to the prediction of ligand binding residues?

Author Response

We upload the response as a file named 'Response to Reviewer4.pdf'

Round 2

Reviewer 2 Report

After revision, the authors improved the general quality of the work by including new discussions and clarifying some of the drawbacks present in the previous version of the manuscript.

The server is now functional. However, the output from the current server prediction is poor. This should be improved in future versions. 

Author Response

Response to Reviewer 2 Comments

We appreciate you for your second round review and generous comment concerning our manuscript entitled “MPLs-Pred: Predicting Membrane Protein-Ligand Binding Sites Using Hybrid Sequence-based Features and ligand-specific models”. The comment is valuable and helpful for revising and improving our paper and research. The Response for the Comment is as following:

Point: After revision, the authors improved the general quality of the work by including new discussions and clarifying some of the drawbacks present in the previous version of the manuscript.

The server is now functional. However, the output from the current server prediction is poor. This should be improved in future versions. 

Response: Thank you for your advice. We will continue to upgrade our web server to make it more friendly to users.

Author Response

Response to Reviewer 3 Comments

We appreciate you for your second round review concerning our manuscript entitled “MPLs-Pred: Predicting Membrane Protein-Ligand Binding Sites Using Hybrid Sequence-based Features and ligand-specific models”.

Since you didn’t make any comments in the second round review, we don’t have to make changes. Thank you for your recognition of our work.

Reviewer 4 Report

The authors have tried to address my concerns    

Author Response

Response to Reviewer 4 Comments

We appreciate you for your second round review concerning our manuscript entitled “MPLs-Pred: Predicting Membrane Protein-Ligand Binding Sites Using Hybrid Sequence-based Features and ligand-specific models”.

Point: The authors have tried to address my concerns

Response: Since you didn’t make any comments in the second round review, we don’t have to make changes. Thank you for your recognition of our work.